# Brain Endothelial Cells in Blood–Brain Barrier Regulation and Neurological Therapy

**DOI:** 10.3390/ijms26125843

**Published:** 2025-06-18

**Authors:** Yuqing Xiang, Qiuxiang Gu, Dong Liu

**Affiliations:** Nantong Laboratory of Development and Diseases, School of Life Sciences, Co-Innovation Center of Neuroregeneration, Nantong University, Nantong 226001, China; 2209310007@stmail.ntu.edu.cn

**Keywords:** blood–brain barrier, brain endothelial cells, signaling pathways, neurological disorders, therapeutic strategies

## Abstract

Brain endothelial cells (BECs) constitute the core component of the blood–brain barrier (BBB), regulating substance exchange between blood and the brain parenchyma to maintain central nervous system homeostasis. In pathological states, the BBB exhibits the disruption of tight junctions, endothelial cell (EC) damage, and increased permeability, accompanied by neuroinflammation, oxidative stress, and abnormal molecular signaling pathways, leading to neurotoxic effects in the brain parenchyma and exacerbating neurodegeneration and disease progression. This review systematically summarizes the developmental origin, structural characteristics, and pathological mechanisms of BECs in diseases such as Alzheimer’s disease, multiple sclerosis, stroke, and glioblastoma with a particular focus on the regulatory mechanisms of the Wnt/β-catenin and VEGF signaling pathways. By integrating the latest research advances, this review aims to provide a comprehensive perspective for understanding the role of BECs in physiological and pathological states and to provide a theoretical basis for the development of BBB-based therapeutic approaches for neurological diseases.

## 1. Introduction

BECs are the core component of the BBB, which strictly regulates the exchange of substances between the blood circulation and the brain parenchyma. Their hallmark tight junctions (claudin-5/occludin-mediated) establish a paracellular seal with exceptional electrical resistance, while polarized membrane domains integrate nutrient transporters (e.g., GLUT1) and ATP-driven efflux systems (P-glycoprotein) for substrate discrimination [1,2,3]. The absence of fenestrations and suppressed pinocytotic activity reinforces barrier selectivity, synergized by endogenous enzymatic networks (monoamine oxidases) that neutralize bioactive molecules [3]. Immune surveillance modulation occurs through spatiotemporal adhesion molecule expression, and their functional integration within neurovascular units (pericytes/astrocytes) enables dynamic barrier regulation through intercellular crosstalk [4]. The BBB is not only an important barrier for maintaining the homeostasis of the central nervous system (CNS) but also plays a key role in neurovascular coupling, immunomodulation, and metabolic support [5,6]. Emerging single-cell resolution histology coupled with programmable nucleases (CRISPR/Cas9, TALENs, and ZFNs) has elucidated developmental trajectories, phenotypic heterogeneity, and disease-associated molecular cascades in BECs [7]. These tools facilitate the targeted genetic perturbations of BEC-associated genes—ablating tight junction components (claudin-5/occludin) to map paracellular barrier dynamics, engineering transporter profiles (P-gp/GLUT1) to model therapeutic biodistribution, or suppressing adhesion molecule expression (ICAM-1/VCAM-1) to decode leukocyte trafficking dynamics [4]. CRISPR dominates in vitro high-throughput screening due to its multiplexing capacity, while TALEN/ZFN systems remain valuable for in vivo applications requiring minimal off-target effects [4,7]. Studies have shown that abnormalities in BECs function are closely related to a variety of neurological diseases, including Alzheimer’s disease (AD), multiple sclerosis (MS), stroke, glioblastoma (GBM), etc. [6]. In these disorders, BBB disruption or dysfunction frequently induces neuroinflammation, vascular leakage, and neuronal damage, thereby accelerating disease progression.

This paper reviews the developmental origins and morphological structure of BECs and their roles in the formation and maintenance of the BBB, with a focus on the centrality of the Wnt/β-catenin and VEGF signaling pathways in the regulation of BEC function. In addition, this paper summarizes the pathological mechanisms of BECs in diseases, such as AD, MS, stroke, and GBM, and looks forward to therapeutic strategies for neurological diseases caused by BECs. By integrating the latest research advances, this paper aims to provide a comprehensive perspective for understanding the role of BECs in physiological and pathological states and to provide a theoretical basis for the development of BBB-based therapeutic approaches for neurological diseases.

## 2. Cellular Origin of Brain Endothelial Cells

Vascular development initially occurs during the formation of the gastrula, with mesodermal progenitor cells determining EC differentiation [8,9]. In zebrafish [10,11], avian embryos [12,13,14], and mice [15,16,17], ECs originate from the lateral plate mesoderm and the paraxial mesoderm. It has long been assumed that new blood vessels can only sprout from pre-existing ECs through angiogenesis after the original vascular network has reformed [18,19]. However, a recent study has identified yolk sac-derived erythro-myeloid progenitors (EMPs) as an alternative source of ECs in the mouse brain [20], which questions the current conventional wisdom that embryonic vasculature relies solely on the proliferation of pre-existing ECs for its expansion. This finding remains controversial, as another study utilizing similar mouse genetics did not confirm that EMPs contribute to the formation of cerebral vascular structures [21]. So it is unclear whether ECs from different embryonic sources have different functions or whether they preferentially promote different cerebral vascular beds.

Previous studies in zebrafish (*Tg(flk1:EGFP)*) have shown that ECs in the head are derived from the anterior lateral plate mesoderm (ALPM). In contrast, ECs in the trunk and tail are derived from the posterior lateral plate mesoderm [22,23,24]. A recent study employing light sheet fluorescence microscopy (LSFM) coupled with dual transgenic zebrafish lines (*Tg(H2A.F/Z:EGFP)*; *Tg(kdrl:mCherry)*) for high-resolution live imaging, and retrospective lineage tracing has demonstrated that the dorsal anterior (DA) cell population in the gastrula serves as the primary origin of cranial endothelial cells in zebrafish, predating the formation of the anterior lateral plate mesoderm (ALPM) [23]. Through innovative AFEIO image processing algorithms and single-cell tracking analysis using Imaris software (version 9.0.1), this work provides the first evidence that endothelial progenitors are broadly distributed along both the dorsoventral (DV) and anteroposterior (AP) axes during gastrulation. These findings were further validated using photoactivatable Kaede protein labeling (*Tg(kdrl:kaede)*), which confirmed the fate commitment of DA progenitors to cranial vasculature.

Despite the current research breakthroughs, the mechanism of distinguishing different vascular EC subtypes in the brain and their genealogical development from the embryonic stage remains a mystery. To fill this knowledge gap, genealogical tracing, clonal analysis, real-time cell tracking, and genomics technologies are needed to map the fate of vascular ECs in the whole brain at single-cell resolution, thus revealing the genealogical development pathway of vascular ECs from embryonic origins to the mature stage.

## 3. Morphology and Structure of Brain Endothelial Cells

The discovery of BBB traces back to the seminal work of Paul Ehrlich [25]. In his landmark experiment with intravenous chondroitin sulfate injection in rodents, he first observed dye penetration in all organs except the brain, laying the experimental foundation for the BBB concept. Modern research has expanded this definition from a simple anatomical separation to a complex interface system encompassing physical barrier properties, selective transport mechanisms, and biochemical regulatory networks.

At the molecular level, the BBB function is primarily mediated by tight junctions between BECs. Claudin-5 (Cldn5), a core member of the claudin family [26,27], is highly conserved across vertebrates [27,28,29,30] and specifically localized at endothelial membrane junctions, serving as a structural hallmark of the BBB [31]. In stark contrast to barrier-forming ECs, fenestrated vascular ECs show negligible Cldn5 expression [29,32,33,34,35] but are enriched with plasmalemma vesicle-associated protein (PLVAP) [36,37,38]. As a key component of fenestral diaphragms, PLVAP not only serves as a molecular marker for fenestrated endothelia [32,34,39,40] but also regulates vascular permeability and immune cell trafficking in specialized brain regions like the choroid plexus [36,37,40].

Notably, the β-catenin signaling pathway maintains BBB integrity by suppressing PLVAP expression [33,34,41]. Tracer experiments combined with magnetic resonance imaging have demonstrated that claudin-5-deficient mice exhibit increased BBB permeability selectively to small molecules (<800 Da) while maintaining effective restriction against larger molecules [42]. These findings reveal that Cldn5 deletion specifically impairs the BBB’s molecular size-selective barrier function, resulting in abnormal small-molecule leakage without affecting macromolecular containment. In mice overexpressing the amyloid β-peptide (Aβ) precursor protein, GLUT1 deficiency triggers early cerebral microvascular degeneration accompanied by reduced blood flow, dysregulated vascular homeostasis, and BBB breakdown [43,44]. This endothelial dysfunction accelerates Aβ pathology through impaired clearance mechanisms while concurrently diminishing neuronal activity and inducing behavioral deficits. Mechanistically, GLUT1-deficient mice exhibit suppressed endothelial lactate production, leading to pericyte detachment in both retinal and brain vasculatures, which consequently exacerbates BBB permeability [43,44]. These initial cerebrovascular impairments precede and drive subsequent progressive neuronal loss and neurodegeneration. Moreover, several independent studies have shown that PLVAP impairment disrupts fenestrae formation and promotes plasma protein extravasation [40,45,46]. For example, the developmental synchronization of zebrafish (*Tg(l-fabp:DBP-EGFP;kdrl:mCherry-caax)*) PLVAP ortholog expression correlates with hypophyseal vascular permeability to plasma proteins and the formation of a selective barrier separating the pituitary from BBB-protected vasculature [40,45,46]. Real-time imaging in larvae revealed the temporal dynamics of plasma protein transit across hypophyseal endothelial cells. Crucially, this study establishes PLVAP as a direct regulator of blood-derived protein transport efficiency through fenestrated endothelial structures into the pituitary [40,45,46]. In PLVAP-deficient mutant mice, the absence of capillary diaphragms disrupts the barrier function of fenestrated capillaries, leading to substantial leakage of plasma proteins [40,45,46]. These findings collectively reveal the sophisticated regulatory relationship between BBB molecular composition and function.

### 3.1. Development and Maintenance of the BBB

The Wnt/β-catenin signaling pathway has been identified as a master molecular switch governing cerebral vascular development and BBB formation (Figure 1). The BAT-Gal transgenic reporter mouse model demonstrates its specific activation in central nervous system vascular ECs [41,47,48] while remaining quiescent in peripheral vasculature [48], suggesting its unique role in BBB development.

Recent animal studies utilizing developmental and adult BAT-gal murine models, complemented by zebrafish (*Tg(kdrl:ras-mCherry):Tg(7xTCF-Xia.Siam:EGFP)*) investigations, reveal substantially lower β-catenin activity in the fenestrated endothelia of choroid plexuses (CPs) and circumventricular organs (CVOs) compared to BBB-forming vasculature [33,34]. Genetic manipulation experiments in BAT-Gal mice establish that (1) endothelial-specific β-catenin knockout during development dramatically reduces BBB marker proteins (e.g., Claudin 5 and GLUT1) while upregulating the permeability marker PLVAP [41,47,48]. (2) Conversely, constitutive β-catenin activation produces opposite effects [41]. Notably, endothelial-specific β-catenin activation in BAT-Gal mice induces the partial conversion of BBB-like barrier properties in normally permeable vasculature, characterized by increased Cldn5-positive vessels with reinforced intracellular junctions and the concomitant suppression of Plvap/Meca32 expression and endothelial fenestration [33]. However, this β-catenin-mediated barrier transformation cannot be replicated in short-term in vitro cultures of mouse CNS endothelia [50], highlighting the crucial role of microenvironmental signals in BBB maintenance.

In mature BBB, β-catenin continues to play indispensable roles: its postnatal deletion in mouse endothelia compromises barrier integrity [51,52], and mechanistic studies show that it regulates pericyte coverage via Pdgfb expression to maintain vascular stability [53]. These findings collectively establish β-catenin’s dual regulatory mechanism, serving as both the molecular foundation for BBB properties and a critical switch that suppresses PLVAP-mediated hyperpermeability phenotypes.

### 3.2. Development and Maintenance of Hypertonic Vasculature

The VEGF signaling pathway has been identified as a central molecular mechanism regulating fenestrated endothelial differentiation. This pathway exerts dual regulatory effects by simultaneously downregulating tight junction proteins (claudin-5 and occludin) while upregulating the key fenestrae-forming protein PLVAP [54,55,56,57]. Notably, remarkable regional heterogeneity exists in VEGF-mediated regulation across different brain areas.

During fasting in male mice (C57Bl/6), decreased blood glucose levels alter the structural organization of the blood–hypothalamic barrier, enhancing metabolic substrate accessibility to the arcuate nucleus. In the median eminence (ME), fasting mice stimulate tanycytes to upregulate Vegf-A expression, consequently enhancing endothelial PLVAP expression and fenestration density [58]. Particularly, in murine models, chemogenetic activation of neurons expressing melanin-concentrating hormone (MCH neurons) or optogenetic stimulation of their ME projections elevates fenestrated vascular loop density in the ME, thereby enhancing its permeability. Subsequent unbiased phosphoRiboTrap-based profiling of neuronal activation states revealed MCH neurons’ capacity to modulate endothelial function. Mechanistically, MCH-expressing neurons regulate localized vascular fenestration patterns and endothelial cell density through axonal Vegf-A secretion [59], a finding that perfectly corroborates VEGF’s central role in vascular remodeling [60].

Studies in the neurohypophysis (NH) reveal that pituitary-derived Vegf-A not only maintains endothelial proliferation in adult male mice (C57BL/6J) [61] but also cooperates with TGF-β to induce *plvap* expression during zebrafish (*Tg(oxt:EGFP): Tg(kdrl:mCherry-caax)*) development, collectively shaping fenestrated vascular characteristics [62]. A transgenic system was engineered for simultaneous in vivo visualization of hypothalamic axons and neurohypophyseal vasculature. Importantly, the integrity of the hypothalamic–pituitary axonal tracts is crucial for this process, as their disruption significantly impairs fenestrated vessel formation [63].

Murine (C57BL/6) studies demonstrate that the choroid plexus (CP) ependymal epithelium maintains capillary fenestration integrity through Vegf-A/TGF-β signaling [55,64]. Recent studies further identified that developing fenestrated vessels in the zebrafish (*Et(cp:EGFP)*) hindbrain CP requires precise combinations of specific Vegf ligands [32], suggesting potential functional specialization among VEGF family members.

In adult male mice (C57BL/6J) the circumventricular organs’, including the area postrema (AP), the organum vasculosum of the lamina terminalis (OVLT), and the subfornical organ (SFO), neuron- and astrocyte-derived VEGF-A continuously sustains fenestrated capillary properties [65]. Interestingly, Wnt inhibitory factor-1 deletion in the AP elevates GLUT1 expression in mouse models [34], providing direct evidence for Wnt/β-catenin signaling’s inhibitory role in BBB formation. Although pineal gland (PG)-derived melatonin has been implicated in angiogenesis regulation [66], its specific role in fenestrated capillary development awaits further clarification.

## 4. Brain Endothelial Cells and Neurological Disorders

### 4.1. Alzheimer’s Disease

Alzheimer’s disease (AD) is characterized not only by classical pathological hallmarks—including amyloid plaques, hyperphosphorylated tau tangles, and neuronal loss—but also by early cerebrovascular dysfunction, which is increasingly recognized as a critical contributor to disease pathogenesis and cognitive decline [67,68,69]. Notably, vascular dysfunction may be one of the earliest detectable biomarker changes in the preclinical phase, preceding clinical symptom onset and changes in conventional AD biomarkers. For example, Iturria-Medina, Y. et al. [70] analyzed more than 7700 brain images from the ADNI and dozens of plasma and cerebrospinal fluid biomarkers to find amyloid deposition, cerebrospinal fluid Aβ42, and tau pathology (pTau and total tau) [68,71]. At the molecular level, AD brains exhibit reduced nuclear localization of β-catenin, leading to decreased expression of its downstream neuroprotective target genes [72,73]. This deficiency is exacerbated by enhanced proteasomal degradation of β-catenin (e.g., via RHBDL4-mediated aberrant processing), which directly correlates with cognitive impairment [72].

A key mechanism linking vascular dysfunction to AD progression is BBB impairment, which disrupts Aβ clearance from the brain to peripheral circulation. This dysfunction is driven by an imbalance in transport receptors at the vascular interface, specifically reduced LRP-1 (low-density lipoprotein receptor-related protein 1) and elevated RAGE (receptor for advanced glycation end products) expression [74].

Central to this process are BECs, which actively participate in AD pathology. Accumulating evidence highlights the pivotal role of VEGF signaling in BECs during AD progression. For instance, the elevated expression of VEGF receptors (e.g., FLT1 and FLT4) in BECs correlates strongly with AD severity: high FLT1 levels are associated with increased Aβ burden and worse cognitive performance, while FLT4 upregulation is linked to Aβ accumulation [54,62]. Importantly, FLT1 is significantly upregulated in AD patients, further implicating BEC-specific VEGF signaling in disease progression [75]. These findings collectively suggest that BECs contribute to AD pathogenesis via dysregulated VEGF pathways, offering a promising therapeutic target.

### 4.2. Multiple Sclerosis

In acute and chronic demyelinating lesions of MS and experimental autoimmune encephalomyelitis (EAE), the distinguishing features are newly formed leaky blood vessels and BBB damage. Studies have shown that venous ECs play a key role in these lesions. Single-cell transcriptome analysis and in vivo validation experiments have shown that venous ECs exhibit neovasculogenesis-related gene signatures and increased proliferation in acute and chronic EAE lesions, leading to venous dilatation and increased venous coverage [76]. These changes are associated with the upregulation of the vascular endothelial growth factor A (VEGF-A) signaling pathway. Studies have also demonstrated increased expression of markers of neovascularization in acute and chronic lesions of human MS. Although the use of a VEGF-A blocking antibody reduced the transcriptional signature of neoangiogenesis and vascular proliferation in EAE mice, it failed to restore BBB function or improve EAE pathology [54]. These findings suggest that venous ECs play an important role in neoangiogenesis under demyelinating neuroinflammatory conditions and provide new insights into understanding the mechanisms of vascular pathology in MS and EAE.

Another study reported that in early MS, inflammatory damage to the BBB is accompanied by the infiltration of pathogenic immune cells into the CNS [77]. Notably, in the EAE model (a key MS animal model), the activation of β-catenin signaling was shown to alleviate neuroinflammation and promote remyelination by suppressing the hyperactivation of the NF-κB and STAT3 pathways [78]. Furthermore, the nuclear translocation of β-catenin inhibits pro-inflammatory phenotypes in microglia and astrocytes, thereby reducing neuroinflammatory damage [78,79]. As MS progresses, dysregulated neurovascular coupling becomes associated with gray matter atrophy in late-stage disease [80,81].

Genetic and environmental factors associated with MS, including dietary habits, gut microbiota, and vitamin D concentrations, may contribute directly or indirectly to brain EC dysfunction [82]. Brain EC damage leads to the entry of harmful molecules into the CNS and accelerated BBB leakage [83]. Potential future therapies may help prevent BBB damage (e.g., monoclonal antibodies targeting cell adhesion molecules and fibrinogen) and repair BBB dysfunction (e.g., mesenchymal stromal cells). These findings provide important clues for understanding the pathomechanisms of MS and developing new therapeutic strategies.

### 4.3. Stroke and Vascular Lesions

Stroke, the second leading global cause of death, imposes significant morbidity, mortality, and disability, with profound socioeconomic consequences [84]. The two primary stroke subtypes—ischemic (87% of cases) and hemorrhagic—differ in etiology. Acute ischemic stroke (AIS) results from cerebral arterial occlusion, abruptly depriving local tissue of oxygen and blood flow [85]. A key pathophysiological mechanism in AIS is BBB disruption, driven by tight junction degradation and heightened endothelial vesicular transport. This breach permits the uncontrolled infiltration of blood components, exacerbating cytotoxic/vasogenic edema and increasing hemorrhagic transformation (HT) risk [86,87,88]. Interestingly, the VEGF and Wnt/β-catenin pathways exhibit antagonistic roles in post-stroke angiogenesis—VEGF inhibition elevates Wnt/β-catenin transcriptional activity [89], suggesting pathway interplay during vascular repair. Clinically, severe BBB disruption correlates with poorer NIH Stroke Scale (NIHSS) scores, worse functional recovery, and higher mortality [90,91], underscoring BBB protection as a therapeutic target to improve outcomes [92].

After stroke, sustained activation of ECs leads to the progression of systemic atherosclerosis, thereby increasing the risk of recurrent vascular events. It has been shown that cerebral ischemia induces the sustained activation of peripheral EC, the upregulation of the adhesion molecule VCAM1, and increased senescence, and these changes persist up to 4 weeks after stroke. This aberrant EC activity is driven by sustained Notch1 signaling, and the increase in Notch1 signaling is triggered by an increase in the circulating Notch1 ligands DLL1 and Jagged1 in mice and humans after stroke [93]. This leads to increased myeloid cell adhesion and promotes atherosclerosis progression by generating senescent pro-inflammatory endothelium. The use of Notch1 or VCAM1 blocking antibodies as well as the knockdown of endothelial Notch1 both reduced atherosclerosis progression after stroke. These findings reveal a systemic mechanism of persistent peripheral EC activation after stroke and provide new ideas for the prevention and therapeutic intervention of recurrent vascular events after stroke.

Cerebral small vessel disease (SVD) is a vascular disease that increases the risk of stroke and dementia and is usually diagnosed by brain MRI. Current primary prevention and secondary treatment of SVD focuses on lifestyle interventions and vascular risk factor control, including blood pressure reduction. However, these interventions have limited effectiveness; some patients with sporadic SVD do not have hypertension, and SVD shows a strong familial and genetic basis. There is growing evidence that brain EC dysfunction is a key mechanism of SVD [80]. Dysfunctional ECs can lead to cerebrovascular dysfunction, alter BBB integrity, and interfere with intercellular interactions in the neuroglial-vascular unit, resulting in damage to neighboring brain tissues. ECs in SVD may become dysfunctional either through intrinsic mechanisms (e.g., genetic predisposition to SVD) or extrinsic factors (e.g., high blood pressure, cigarette smoking, and diabetes). Drugs targeting the endothelial pathway have shown promise in clinical trials, and understanding their effects on ECs and the surrounding brain tissue may help in the development of additional therapies to limit disease progression and improve the prognosis of patients with SVD [94].

### 4.4. Glioblastoma

Glioblastoma (GBM), the most aggressive and treatment-resistant primary brain tumor, is characterized by diffuse infiltration, frequent recurrence, and poor clinical outcomes despite standard therapies [95]. Notably, there are reports that the BBB in GBM can remain intact, and this preserved barrier structure actively impedes the delivery of therapeutic agents to tumor sites [96]. At the molecular level, the hyperactivation of the Wnt/β-catenin signaling pathway is closely associated with GBM stem cell (GSC) self-renewal, tumor invasiveness, and therapy resistance [97,98]. This oncogenic pathway drives tumor progression through β-catenin-mediated upregulation of downstream targets (LEF1 and c-Myc) that promote proliferation and angiogenesis [99]. The angiogenic switch in GBM is particularly dependent on VEGF-A, which activates VEGFR signaling to stimulate endothelial proliferation and neovascularization, thereby sustaining tumor growth [100,101].

Recently, Yuan Xie et al. performed single-cell RNA sequencing on human cerebrovascular cells from 13 surgically resected GBM samples and adjacent normal brain tissues [102]. They identified a series of transcriptomic abnormalities in GBM and determined that LOXL2-dependent collagen modification is a common GBM-specific feature. Furthermore, they demonstrated that LOXL2 inhibition enhances chemotherapeutic efficacy in both mouse and patient-derived xenograft (PDX) GBM models.

Cancer immunity is spatiotemporally regulated by leukocyte interactions with tumor cells and stromal cells, leading to immune escape and immunotherapy resistance. A distinct population of mesenchymal-like ECs was found to exist in GBM, forming an immunosuppressive vascular microenvironment [103]. Mechanistically, studies have revealed a spatially restricted Twist1/SATB1-mediated sequential transcriptional activation mechanism through which tumor ECs produce osteoblasts and promote an immunosuppressive macrophage (Mφ) phenotype [104]. Genetic or pharmacological elimination of Twist1 reverses Mφ-mediated immunosuppression and enhances T-cell infiltration and activation, thereby reducing GBM growth, prolonging mouse survival, and sensitizing tumors to chimeric antigen receptor T-cell immunotherapy [105]. These findings reveal spatially restricted mechanisms that control tumor immunity, suggesting that targeting endothelial Twist1 may offer promising prospects for optimizing cancer immunotherapy in combination with existing anti-angiogenic approaches.

## 5. Therapeutic Strategies for Targeting Brain Endothelial Cells

### 5.1. Repair and Protection of BBB Integrity

With more and more people in the world suffering from central nervous system diseases, effective strategies are urgently needed to develop targeted therapy [106]. Since the BBB effectively blocks the delivery of most neurotherapeutic drugs to the brain, how to cross the BBB is an important issue to consider when developing targeted drugs for the treatment of neurological disorders. At the same time, the disruption of BBB integrity is a common pathological feature of many CNS disorders, and therefore, therapeutic approaches targeting the BBB itself need to be explored and refined [107].

#### 5.1.1. Wnt/β-Catenin Pathway Agonists

Since Wnt/β-catenin signaling plays an important role in the maintenance of the BBB, therapeutic approaches targeting this pathway can be instrumental in repairing the BBB. However, the Wnt signaling pathway exhibits remarkable tissue-specific regulatory heterogeneity. Studies have demonstrated that pathway activation can effectively enhance tight junctions and restore barrier function in brain microvascular endothelial cells and retinal endothelial cells while paradoxically increasing permeability in choroidal endothelial cells [108]. This dual, tissue-dependent effect poses significant challenges for systemic administration, potentially causing barrier dysfunction in non-target tissues. Furthermore, native Wnt ligands (e.g., Wnt7a and Norrin) present substantial druggability obstacles: these proteins require complex post-translational lipid modifications (e.g., palmitoylation) for functionality and depend on specific coreceptor complexes (e.g., Frizzled/LRP) to initiate downstream signaling [109,110]. These characteristics result in the poor solubility and low stability of native ligands, severely limiting their potential as therapeutic agents.

Martin et al. identified a class of Gpr124/Reck agonists by leveraging the specificity of the Wnt7a/b-Gpr124/Reck co-receptor complex, which differs from Wnt7a by only a single surface-exposed residue [111]. Unlike Wnt ligand overexpression, which is incompatible with vertebrate development, Gpr124/Reck agonists are well tolerated in vivo and are delivered normally in African clawed toad, zebrafish, or neonatal mouse brains (Figure 2). At the same time, the agonist showed promising therapeutic effects in mouse models of brain tumors and ischemic stroke, and intravesical gene delivery achieved durable normalization of the BBB. Therefore, normalizing the BBB by restoring Wnt signaling in ECs could be a good approach to treating the BBB. Nevertheless, future studies must focus on enhancing agonist specificity to target tissues while avoiding off-target effects on developmental processes [112,113].

#### 5.1.2. Anti-Inflammatory Factors

Inflammatory factors such as TNF-α and IL-1β disrupt the tight junctions of BBB by activating the NF-κB signaling pathway. For example, the activation of the TLR4/NF-κB signaling pathway directly impairs the integrity of the BBB by downregulating the expression of proteins such as ZO-1, occludin, etc. [114,115,116]. Inflammatory signaling will further exacerbate NF-κB activity by activating oxidative stress pathways, creating a vicious cycle. For example, microcystin exposure exacerbates neuroinflammation and BBB leakage by activating the NLRP3 inflammatory vesicle, leading to decreased expression of the BBB’s tight junction proteins occludin and cldn5, as well as the release of the pro-inflammatory factor S100B.

Therefore, it can protect the BBB by inhibiting the Nf-κB signaling pathway. Metformin reduces the release of pro-inflammatory factors (TNF-α, IL-1β, and IL-6) by activating AMPK signaling and inhibiting the nuclear translocation of NF-κB. For example, in a sepsis model, metformin significantly inhibited NF-κB activity while upregulating occludin and claudin-3 expression, improving BBB function and attenuating brain damage [117,118,119]. At the same time, metformin also inhibits oxidative stress and mitochondrial dysfunction by activating the AMPK pathway, thereby reducing inflammatory damage to the BBB (Figure 2). For example, in a traumatic brain injury model, metformin attenuates increased BBB permeability and secondary injury by compensating for defective energy metabolism [119,120,121]. In addition to the direct regulation of tight junction proteins, metformin attenuates the impact of systemic inflammation on the BBB through indirect pathways such as the inhibition of STAT3 acetylation, the modulation of intestinal flora, and short-chain fatty acid metabolism [122,123].

While pathway inhibition can ameliorate inflammation-mediated barrier disruption, the ubiquitous physiological roles of NF-κB raise concerns about potential systemic immunosuppression and off-target effects [124,125]. Agents like metformin demonstrate pleiotropic neuroprotective effects through AMPK activation, NF-κB suppression, and mitochondrial stabilization, yet achieving endothelial-specific modulation without compromising other cellular functions remains technically challenging [124,125]. Current pharmacological approaches face significant limitations, including poor BBB penetration of small-molecule inhibitors and dose-dependent metabolic complications [126,127]. The intricate crosstalk between NF-κB, NLRP3 inflammasome, and STAT3 signaling pathways further underscores the need for integrated therapeutic strategies rather than single-pathway interventions [127,128,129]. Emerging solutions include smart nanocarriers with microenvironment-responsive drug release profiles and rationally designed combination therapies that address the multifactorial nature of neuroinflammation while preserving systemic immune homeostasis [130].

#### 5.1.3. Anti-Oxidative Stress

Mitochondrial dysfunction in BECs is the core mechanism of BBB injury, and the core of targeted mitochondrial function therapy lies in the modulation of processes such as mitochondrial dynamics, mitochondrial autophagy, and biosynthesis to restore mitochondrial function [131]. Studies have shown that abnormal mitochondrial function is closely associated with a variety of diseases, including cardiovascular disease, neurodegenerative disease, and cancer [132,133]. For example, in AD, the accumulation of amyloid precursor protein (APP) in mitochondria is significantly correlated with the decline of mitochondrial function, suggesting that targeting mitochondrial APP may be a new direction for the treatment of AD [134,135]. In addition, the accurate measurement of mitochondrial DNA copy number (mtDNAcn) could help to identify cancers that depend on enhanced mitochondrial function, thus providing a basis for the application of mitochondrial inhibitors [136]. In terms of therapeutic strategies, mitochondria-targeted nanomedicines show great potential in their ability to overcome the delivery barriers of conventional drugs and precisely modulate mitochondrial function [137,138]. For example, mitochondria-targeted nanoparticles (NPs) not only reduce reactive oxygen species (ROS) production but also improve mitochondrial respiratory function and can effectively repair mitochondrial dynamics and biosynthesis at lower concentrations [139]. Chunhong Gao et al. developed an innovative biomimetic nanoplatform by engineering red blood cell (RBC) membrane-coated nanoparticles conjugated with both T807 (a brain-targeting ligand) and triphenylphosphine (TPP, a mitochondrial-targeting moiety) for blood–brain barrier (BBB) penetration and neuronal mitochondria-specific delivery of therapeutic antioxidants [140]. Their results demonstrated that this T807/TPP-RBC-NP system effectively alleviated AD pathology by reducing mitochondrial oxidative stress and preventing neuronal apoptosis in both cellular and animal models (Figure 2). Mitochondrial transplantation techniques have also shown significant results in acute inflammation models, such as the L6 cell-derived mitochondrial transplantation that significantly enhanced mitochondrial function and attenuated inflammatory responses [141].

Mitochondrial dysfunction in BECs has emerged as a pivotal factor in BBB disruption, positioning mitochondrial homeostasis restoration through the modulation of dynamics, mitophagy, and biogenesis as a promising therapeutic avenue [142,143]. While mitochondrial-targeted nanoparticles demonstrate considerable potential in enhancing drug delivery and improving oxidative phosphorylation, their clinical application is constrained by the inherent challenges of achieving precise cerebral endothelial targeting given the BBB’s selective permeability [139,144]. The emerging technique of mitochondrial transplantation, despite showing efficacy in acute injury models, faces unresolved questions regarding donor cell compatibility, long-term functional integration, and the standardization of engraftment protocols [145]. Furthermore, therapeutic strategies targeting mitochondrial components must carefully consider the delicate balance between desired effects and potential systemic consequences due to the fundamental role of mitochondria across all cell types. The translation of these approaches is further complicated by the current lack of reliable delivery systems capable of ensuring both targeted action and immunological safety, particularly concerning potential immune reactions to exogenous mitochondria or nanocarrier components. Addressing these multifaceted challenges will necessitate the development of innovative cell-specific delivery platforms combined with integrated treatment strategies that simultaneously target mitochondrial dysfunction and its downstream effects on BBB integrity while maintaining rigorous safety profiles for clinical application.

### 5.2. Enhanced BBB Penetration (Drug Delivery Strategy)

#### 5.2.1. Carrier-Mediated Delivery Systems

##### Nanoparticles (NPs)

Nanoparticles have emerged as promising vehicles for drug delivery across the BBB, with lipid-based systems—including liposomes, solid lipid nanoparticles (SLNs), and nanostructured lipid carriers (NLCs)—demonstrating particular advantages due to their phospholipid bilayer structure that mimics BBB membrane composition, thereby enhancing drug encapsulation and trans-barrier transport [146,147,148]. Beyond serving as protective carriers for hydrophobic drugs [149], SLNs and NLCs can achieve brain-targeted delivery through receptor-mediated transcytosis and other active transport mechanisms [150]. To further enhance targeting specificity, surface functionalization with ligands (e.g., lactoferrin, polyethylene glycol [PEG], and transferrin) enables receptor-mediated transcytosis—transferrin-conjugated liposomes, for instance, efficiently traverse the BBB via transferrin receptor binding [151,152,153]. Dual-ligand systems (e.g., lactoferrin combined with muscone) demonstrate synergistic targeting effects [154], while matrix metalloproteinase (MMP)-responsive liposomes achieve stimuli-triggered drug release in disease-specific microenvironments [155].

Polymeric nanoparticles, especially those fabricated from poly (lactic-co-glycolic acid) (PLGA), offer additional benefits through tunable degradation rates and high drug-loading capacity, showing excellent BBB penetration in models of GBM and neurodegenerative diseases [120,156,157]. Surface engineering strategies further optimize delivery: PEGylation prolongs systemic circulation while reducing immune clearance [120,158], though the impact of the PEG molecular weight (2–10 kDa) on BBB penetration requires further investigation [158]. Targeted functionalization with ligands like the angiopep-2 peptide significantly enhances BBB traversing capability [158].

Despite significant progress in nanoparticle-mediated drug delivery to the brain, several critical challenges hinder clinical translation. While PEGylation effectively prolongs systemic circulation and reduces immune clearance, it may trigger anti-PEG antibody responses, leading to accelerated blood clearance (ABC phenomenon) and complement activation-related toxicity. These effects are particularly pronounced upon repeated administration, substantially compromising delivery efficiency [159,160].

Furthermore, the immunogenic potential of nanocarriers remains a major concern. Lipid-based nanoparticles (e.g., SLNs/NLCs) can activate innate immune responses through their phospholipid components, while degradation byproducts of polymeric nanoparticles (e.g., PLGA) may induce local inflammatory reactions. Emerging mitigation strategies include optimizing material biocompatibility through chemical inertness or incorporating immune-shielding coatings such as polysialic acid [161,162].

##### Exosome Engineering

Exosomes have emerged as a breakthrough biological delivery system for central nervous system disorders, leveraging their innate ability to traverse the BBB while avoiding immune rejection. These natural nanovesicles can directly penetrate into the brain parenchyma through receptor-mediated transcytosis [163,164,165], offering unparalleled advantages including inherent low immunogenicity [166,167], exceptional BBB-penetrating capability [166], and intrinsic biocompatibility. Their unique properties make them ideal carriers for protecting vulnerable payloads like siRNA from degradation while enabling brain-specific delivery.

The therapeutic potential of exosomes can be further enhanced through surface engineering strategies. By incorporating targeting peptides or antibodies, engineered exosomes demonstrate improved specificity for neurons or BBB components [168]. For instance, in GBM treatment, folate-modified exosomes co-loaded with temozolomide (TMZ) and quercetin have shown remarkable efficacy in simultaneously overcoming TMZ resistance (mediated by MGMT overexpression) while reducing systemic toxicity [169,170]. This approach addresses key limitations of conventional chemotherapy where TMZ, despite its BBB permeability, often shows restricted efficacy due to drug resistance and adverse effects [171,172].

Beyond small molecules, exosomes have proven versatile in delivering diverse therapeutics including chemotherapeutic agents, nucleic acids, and neurotrophic factors [173,174]. Various loading techniques such as electroporation and sonication have been successfully employed, with demonstrated therapeutic effects in Parkinson’s disease models through the precise modulation of neuroinflammatory pathways [175]. The platform’s flexibility is further exemplified by dual-targeting systems against GBM [176,177] and optimized transcytosis mechanisms through endocytosis/exocytosis regulation [174,178].

While exosome technology shows tremendous promise, several translational challenges must be addressed. Engineering modifications may affect the natural low immunogenicity of exosomes, and targeting efficiency varies significantly depending on cellular origin (e.g., microglia vs. astrocyte-derived exosomes). Other limitations include suboptimal drug loading efficiency (<30%), manufacturing heterogeneity, and insufficient long-term safety data regarding BBB integrity. Future research should focus on donor cell selection, modular engineering platforms, and clinical-grade production optimization to realize the full potential of this innovative delivery system [173,179].

#### 5.2.2. Physical/Chemical Methods for the Instantaneous Opening of the BBB

##### Focused Ultrasound Combined with Microbubbles

The BBB can be reversibly opened locally using ultrasound-mediated microbubble oscillations (focused ultrasound combined with microbubbles.) Clinical trials of opening the BBB in multifocal brain regions have shown that the treatment procedure did not cause serious adverse effects. Preliminary studies in AD patients confirmed the safety of the treatment by monitoring physiologic indices (e.g., EEG and hemodynamic parameters) [180]. Ultrasound-mediated BBB opening enhances the delivery efficiency of other therapies (e.g., anti-Aβ antibodies and neurotrophic factors). For example, microbubble-enhanced FUS successfully delivered anti-Aβ antibodies non-invasively into the brains of AD mice without increased toxicity [181]. Similar strategies may improve drug efficacy and reduce dose-related side effects in clinical trials [182].

The focused ultrasound combined with microbubbles (FUS + MBs) technique offers an innovative solution for transient BBB opening, yet its clinical application faces multiple challenges. Regarding safety, studies demonstrate that ultrasound parameters and microbubble characteristics may induce sterile inflammatory responses, manifested by the upregulation of pro-inflammatory cytokines such as IL-1β, indicating the need for precise parameter optimization to balance BBB opening efficacy with immune activation risks [183,184]. While MRI-guided FUS systems have achieved millimeter-scale targeting precision [185,186], skull attenuation effects and individual anatomical variations may still compromise targeting efficiency, prompting researchers to develop improved solutions like bimodal ultrasound arrays and perfluoropropane-loaded microbubbles [187,188]. For clinical translation, although preliminary trials have confirmed the short-term safety of this technique in neurodegenerative diseases and brain tumor patients [189,190], further validation is still required for long-term repeated treatment safety thresholds, standardized parameter determination, and synergistic effects with systemic therapies [191,192]. Moreover, the sensitivity limitations of current real-time monitoring techniques like indocyanine green methods highlight the necessity to develop more accurate BBB status assessment approaches [193].

##### Mannitol Shrinks Endothelial Cells

Mannitol, as a non-metabolizable hypertonic substance, induces the migration of water molecules from the cell to the outside by increasing the osmotic pressure of the extracellular fluid, resulting in the dehydration and contraction of ECs. This volume change directly affects the mechanical stability of intercellular junctions [194,195]. For this reason, mannitol is widely used in neurosurgery to open BBB, temporarily relaxing tight junctions through the osmotic gradient between vascular ECs, thereby allowing the passage of macromolecular drugs [195,196]. Unlike glucose, mannitol is not metabolized by cells and therefore maintains a more persistent osmotic pressure difference [197,198]

Mannitol-induced BBB opening through endothelial cell contraction for enhanced drug delivery faces several critical challenges. The non-specific osmotic effects may expose healthy brain tissue to potential toxins, while a lack of spatial precision increases off-target risks [191,199]. Chronic or high-dose administration can cause neuronal/astrocytic damage and systemic complications like electrolyte imbalances, adversely affecting postoperative recovery and neural stem cell function [200]. Clinical application remains controversial regarding optimal dosing and safety monitoring, with unresolved debates about its efficacy compared to hypertonic saline in posterior fossa surgery [201]. Traditional administration methods (e.g., intra-arterial infusion) carry risks of visual impairment and cerebral swelling, requiring rigorous fluid management [201]. Furthermore, the transient BBB opening limits therapeutic time windows, and synergistic mechanisms with complementary approaches (e.g., focused ultrasound or nanocarriers) need further investigation [201]. These limitations underscore the urgent need for more precise BBB modulation technologies and personalized treatment protocols.

### 5.3. Cell Therapy and Gene Editing

#### 5.3.1. Transplantation of Endothelial Progenitor Cells (EPCs)

In recent years, endothelial progenitor cells (EPCs) have become a research hotspot in the treatment of ischemic brain injury due to their powerful vascular repair ability. Studies have shown that EPCs can not only differentiate directly into mature ECs but also regulate the neurovascular microenvironment by secreting various active factors, thus promoting brain tissue repair [202].

EPCs are directly involved in the formation of new blood vessels through the secretion of growth factors (e.g., vascular endothelial growth factor (VEGF)) and regulate the activity of the peripheral cells through paracrine signaling to promote endothelial maturation and vascular repair [203]. In a mouse model of middle cerebral artery occlusion (MCAO), the transplantation of EPCs significantly reduced cerebral infarct volume and improved neurological function scores (e.g., the mNSS score) [202,204]. This effect was closely related to the promotion of neovascularization and the inhibition of the inflammatory response by EPCs [202,205]. The restoration of blood flow and improved neuronal survival in the infarcted area were observed by intravenous or intracerebral injection of EPCs or their derivatives (e.g., exosomes) [204,206].

EPC transplantation for ischemic brain injury faces several critical challenges. The therapeutic potential is significantly limited by poor homing efficiency, inadequate migration capacity, and low survival rates of transplanted EPCs, which are closely associated with their functional integration in the ischemic microenvironment [8,207,208,209]. While carotid artery delivery improves cerebral distribution [210], insufficient targeting specificity and BBB penetration remain major obstacles. Clinically, low cell yield, functional decline during expansion, and standardization issues in large-scale production hinder widespread application [209,211]. The angiogenic capacity of transplanted EPCs is further compromised by inflammatory and oxidative stress in the ischemic niche [202,212]. Although synthetic mRNA engineering or combination therapies (e.g., tanshinone IIA) show promise in enhancing EPC function [208,213], their long-term safety and stability require further validation. Current research focuses on optimizing delivery routes, improving cell survival, and developing functionally enhanced EPC subpopulations to increase treatment precision [207,209].

#### 5.3.2. Gene Editing Techniques

##### CRISPR-Cas9 Corrects Transporter Defects

ABCB1, a member of the ATP-binding cassette (ABC) family of transporter proteins, reduces intracellular drug concentrations by actively effluxing chemotherapeutic drugs, leading to multidrug resistance (MDR) in tumors such as GBM. Its high expression in the BBB and tumor cells significantly limits the accumulation and therapeutic efficacy of drugs such as TMZ and carmustine in the brain [214,215,216]. Knockdown of the ABCB1 gene by CRISPR-Cas9 can directly block its mediated drug efflux function and restore intracellular accumulation of chemotherapeutic drugs. For example, the knockdown of ABCB1 in GBM cells significantly enhanced the cytotoxicity of TMZ [214]. Meanwhile, the knockdown of ABCB1 also restored the sensitivity of multidrug-resistant cancer cells to chemotherapeutic drugs, such as paclitaxel and vincristine [217,218,219].

However, CRISPR-Cas9 technology carries inherent limitations that constrain its therapeutic potential. The primary concern remains off-target effects, which may lead to unintended genomic modifications with potential consequences including the dysregulation of critical genes or the activation of oncogenic pathways [220]. Furthermore, the irreversible nature of gene editing raises unresolved questions regarding long-term genomic stability and cellular homeostasis [220]. From a therapeutic perspective, compensatory upregulation of alternative ABC transporters in tumor cells may circumvent ABCB1 knockout, potentially leading to multidrug resistance recurrence [221,222]. These challenges underscore the need for substantial improvements in editing specificity and safety profiles before clinical translation can be realized.

##### AAV Delivery

Viral vectors, particularly adeno-associated viruses (AAVs) and lentiviruses, have emerged as promising platforms for overcoming the BBB in gene therapy, though significant challenges remain. Recent advances in capsid engineering have yielded particularly impressive results. Qin Huang et al. developed BI-hTFR1, an AAV variant that specifically binds to a human transferrin receptor (TFR1) expressed at the BBB [201]. This engineered vector demonstrates remarkable efficiency, showing 40–50 fold greater CNS reporter expression in human TFRC knockin mice compared to conventional AAV9. When delivering the Parkinson’s disease-associated GBA1 gene, BI-hTFR1 significantly enhanced glucocerebrosidase activity in both brain tissue and cerebrospinal fluid, highlighting its potential for human CNS gene therapy applications.

Other engineered AAV variants, including AAV9-derived serotypes such as AAV.CAP-Mac and AAV.CAP-B10, demonstrate enhanced BBB penetration through directed evolution or capsid modifications. Notably, AAV.CAP-Mac was selected via high-throughput screening and exhibits superior BBB traversal compared to wild-type AAV in primates [223], while AAV.CPP.16/21 incorporates targeting peptides to significantly enhance parenchymal transduction efficiency [224]. Additional variants like AAV.PHP.eB have also shown promising systemic administration-mediated brain transduction in non-human primates [225,226].

While lentiviral vectors can bypass the BBB through strategies like ApoEII peptide fusion or intranasal delivery [227,228], their BBB penetration efficiency remains substantially inferior to optimized AAV vectors. This performance gap highlights the need for further engineering of lentiviral systems to improve their CNS delivery capabilities.

Despite considerable progress, viral vector-based gene therapy for central nervous system disorders continues to face several critical barriers. Key limitations include immune-related challenges and insufficient targeting specificity, which may compromise therapeutic efficacy while potentially increasing safety risks [229]. Furthermore, pronounced interspecies variations in BBB penetration efficiency [223,226] and the inherent dose-toxicity dilemma [230] present substantial obstacles for clinical translation.

To address these challenges, next-generation strategies should focus on developing innovative hybrid delivery platforms (e.g., AAV–exosome composite vectors) [231] and exploring alternative administration routes such as intranasal delivery [232]. However, successful clinical implementation will ultimately depend on resolving persistent issues regarding technical complexity and establishing comprehensive long-term safety profiles. Overcoming these barriers represents the crucial next step toward achieving meaningful therapeutic breakthroughs in neurological gene therapy.

Collectively, these multifaceted approaches—from molecular pathway modulation and advanced drug delivery systems to cutting-edge cellular and gene therapies—represent a promising frontier in overcoming BBB challenges, offering new hope for treating neurological disorders by either repairing barrier integrity or circumventing its restrictive nature (Figure 3).

## 6. Future Perspectives

The mechanism of the BBB in neurological diseases and the impact of its changes on the central nervous system microenvironment are the focal points of current research. However, many questions remain unanswered regarding the specific changes in the BBB in disease and its effect on disease progression.

The BBB is not just a single entity that is either “on” or “off”; rather, its complex physiologic functions are affected by various changes in disease. Different studies have often used a single method to detect the integrity of the BBB, such as autopsy tissue analysis, cerebrospinal fluid or blood marker assays, the quantification of exogenous tracer leakage, or in vivo imaging [233]. However, these methods cannot fully reflect the multidimensional changes in the BBB. Future studies need to more precisely resolve the specific changes in the BBB in each disease, especially whether the same or different signals induce BBB disruption in different neurological disorders. If a common mechanism exists, therapeutic strategies may be designed for multiple diseases.

Multiple molecular factors have been shown to modulate BBB dysfunction in various diseases, including VEGF [54,234] and inflammatory factors such as tumor necrosis factor alpha [235], interleukins 1 and 6 [236,237,238], reactive oxygen species [239,240,241], and matrix metalloproteinases [242,243]. However, BBB dysfunction is not solely caused by “disruptive signaling”, it may also be linked to disturbances in maintenance signaling. For instance, the disruption of the Wnt signaling pathway leads to increased vascular permeability and disease progression [244,245], whereas the enhancement of Wnt signaling in CNS ECs may offer therapeutic potential.

Subtle changes in BBB properties can lead to specific neurological symptoms. For example, the dysfunction of multiple BBB transporter proteins has been associated with specific developmental disorders [246], and there may be many more unrecognized similar cases. The regional heterogeneity of the BBB may render specific brain regions more susceptible to certain disease pathologies. If the BBB specializes according to the trophic and signaling needs of specific brain regions, the loss of a particular specialization might result in functional deficits within local neural circuits. In future studies of the neurological therapy targeting the BBB endothelial cells, the focus should be on the temporal–spatial regulation of the aforementioned signaling.

The function of the BBB is not only endothelium-dependent but also closely related to other cell types. The disruption of pericyte coverage results in an increase in nonspecific transcellular transport of ECs and the expression of leukocyte adhesion molecules; however, its specific role in neurological disorders remains unclear. In addition, the disruption of astrocyte end-feet in the neurovascular unit (NVU) decreases clearance by the lymphoid system and may lead to pathological accumulation of proteins such as Aβ. Future studies need to analyze the changes in each cell type, glycocalyx, and basement membrane in the NVU to fully understand the pathophysiological mechanisms of various neurological diseases. Additionally, the crosstalk between these cell types should be emphasized.

Furthermore, the pathological mechanisms of cerebrovascular diseases, particularly ischemic stroke, underscore the essential role of BECs in vascular stability and injury repair. In acute ischemic stroke, early BBB breakdown arises from tight junction degradation, increased endothelial vesicle transport, and the activation of inflammatory signaling pathways, leading to vasogenic edema and neuronal damage [84,85,86].

Notably, VEGF and Wnt/β-catenin signaling exert opposing effects in these pathological processes—VEGF promotes endothelial permeability and angiogenesis, whereas Wnt signaling maintains barrier integrity [88]. These insights suggest that the therapeutic modulation of these pathways may offer unified strategies for stroke intervention. Moreover, chronic endothelial activation post-stroke drives systemic vascular dysfunction and atherosclerosis progression via sustained Notch1 signaling, indicating that anti-senescence or anti-Notch therapies may mitigate long-term complications and recurrent vascular events [92]. Future investigations should delineate the spatiotemporal mechanisms of BEC injury and repair and identify therapeutic strategies to stabilize or reverse BBB dysfunction during cerebrovascular and neurodegenerative diseases.

## Figures and Tables

**Figure 1 ijms-26-05843-f001:**
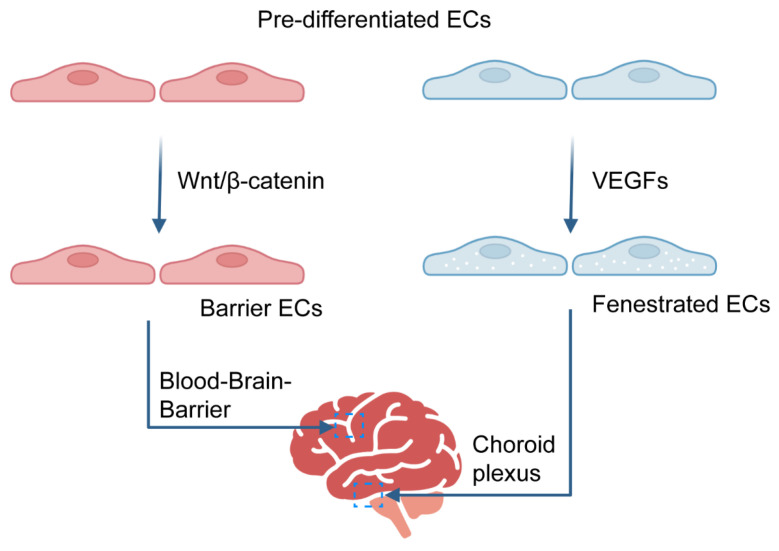
Molecular regulation of BBB and fenestrated vasculature. Schematic representation of two molecular regulation pathways involved in the formation of the BBB and the fenestrated vasculature [49]. Endothelial progenitor cells have the potential to differentiate into distinct endothelial subtypes depending on the microenvironmental cues and signaling pathways. Specifically, they can give rise to (1) barrier-forming brain microvascular ECs that contribute to the BBB, a process predominantly driven by Wnt/β-catenin signaling. Alternatively, (2) under the influence of VEGF signaling, these progenitor cells can differentiate into specialized fenestrated ECs (fenestrated ECs), such as those found in choroid plexus.

**Figure 2 ijms-26-05843-f002:**
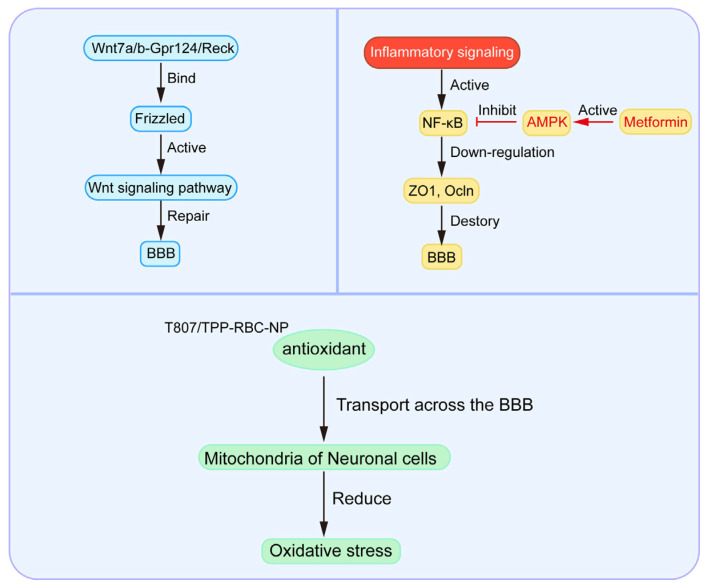
Multi-target therapeutic strategies for BBB repair. Schematic presentation of BBB treatment by targeting the Wnt signaling pathway, anti-inflammatory factors, and mitochondria. (1) The Wnt7a/b—Gpr124/Reck co-receptor complex binds to Frizzled receptors, activating the Wnt signaling pathway to restore BBB integrity. (2) Inflammatory signaling activates NF-κB, leading to the downregulation of tight junction proteins (ZO-1, occludin, etc.) and subsequent BBB impairment. Metformin counteracts this process by suppressing NF-κB activation through AMPK pathway stimulation. (3) Therapeutic antioxidant-loaded T807/TPP-RBC-NPs demonstrate direct mitochondrial targeting capability in neurons, effectively reducing oxidative stress to alleviate pathological symptoms. Future development should focus on engineering nanoparticles specifically targeting endothelial cell mitochondria within the BBB.

**Figure 3 ijms-26-05843-f003:**
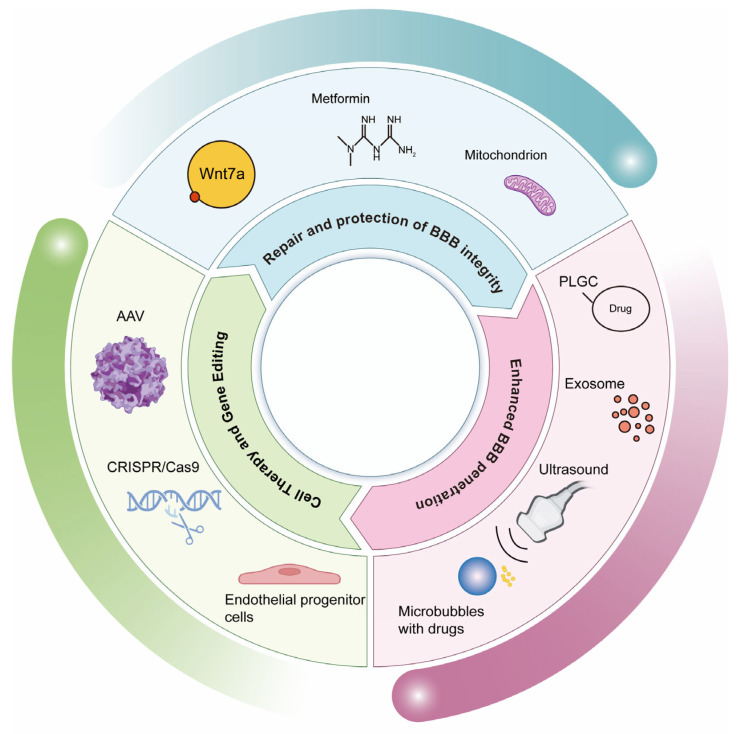
Multifaceted therapeutic strategies targeting the BBB. Schematic representation of therapeutic strategies targeting EC regulation, drug delivery optimization, and genetic interventions, providing precision treatment options for neurological disorders. Current BBB restoration strategies implement synergistic multidimensional approaches: (1) Pathway modulation—Wnt7a/β-catenin activation to enhance tight junction integrity and metformin-mediated AMPK/NF-κB axis inhibition to attenuate inflammation; (2) Mitochondrial remediation—antioxidant therapies combined with metabolic stabilization to address dysfunction; (3) Delivery innovations—engineered nanocarriers (liposomes, PLGA nanoparticles, and TfR-targeted platforms) for BBB-penetrant drug transport, alongside microbubble-assisted focused ultrasound for transient barrier opening validated in preclinical/clinical settings; (4) Cellular/genetic interventions—EPCs for vascular repair and CRISPR-Cas9-driven transporter gene editing to optimize cerebral drug biodistribution and efficacy.

## Data Availability

Not applicable.

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
