# Peer review of "Brain Endothelial Cells in Blood–Brain Barrier Regulation and Neurological Therapy"

_ijms, 2025, doi:10.3390/ijms26125843_

Round 1
Reviewer 1 Report
Comments and Suggestions for Authors
The manuscript by Xiang et al. explores the role of brain endothelial cells in the regulation of the blood–brain barrier (BBB), with a focus on therapeutic strategies targeting these cells in neurological diseases. While the topic is highly relevant and timely, the manuscript suffers from several major issues that need to be addressed. Despite an extensive number of citations, the discussion in many sections remains too superficial, lacking in mechanistic depth and experimental specificity. Furthermore, some statements are presented without appropriate referencing. A major revision is therefore strongly recommended.
Major comments:
Introduction (Lines 33–41):
- In line 33, I recommend the Authors list and briefly describe the key morphological and functional characteristics of brain endothelial cells (e.g., tight junctions, expression of transporters).
-
Lines 36–41 mention gene editing tools, but no specific examples are given. Please specify tools such as CRISPR/Cas9, or TALENs, and/or ZFNs, and briefly explain their relevance to BBB research.
-
Notably, no references are provided in this entire section. Even for introductory content, appropriate citations are essential.
1. Cellular Origin of Brain Endothelial Cells:
-
The section lacks methodological and experimental detail. For instance, when referring to developmental studies, it would be important to specify the use of zebrafish models, and describe the techniques used (e.g., transgenic lines, live imaging etc.).
-
The conclusions presented are not sufficiently supported by detailed evidence. The section would benefit from greater clarity regarding how these insights were obtained.
2. Morphology and Structure of Brain Endothelial Cells:
-
Once again, this section lacks specific references to animal models. A review article should clearly state which systems (e.g., mice, rats, in vitro human-derived cells) underline the presented findings.
3. Brain Endothelial Cells and Neurological Disorders:
-
The discussion of disease models is too limited. The Authors should expand descriptions of individual disorders such as Alzheimer’s disease, gliomas, and stroke, and link them more clearly to BBB dysfunction.
-
Moreover, the Authors should elaborate on the specific studies that support their conclusions, including the models used and methods of evaluating endothelial pathology.
4. Therapeutic Strategies for Targeting Brain Endothelial Cells:
-
This is the most underdeveloped section of the manuscript. The overview of drug delivery methods across the BBB is overly general and lacks a comprehensive analysis of current strategies.
-
The manuscript focuses mostly on PLGA nanoparticles, but omits discussion of other important delivery systems such as:
-
Lipid-based nanoparticles, PEGylated systems, etc.;
-
Viral vectors (e.g., AAV, lentiviruses) commonly used in gene therapy targeting the Central Nervous System;
-
Exosomes, which are increasingly recognized as promising natural carriers due to their biocompatibility and ability to cross the BBB.
-
-
The section should also address the challenges and translational barriers associated with these delivery systems, including immunogenicity, targeting specificity, and clinical feasibility.
Summary:
Although the manuscript tackles an important area of neurovascular research, the current version lacks sufficient scientific depth, methodological precision, and comprehensive coverage of key literature. Major improvements are needed in structure, clarity, and content—particularly regarding therapeutic approaches and mechanistic insights.
Author Response
ijms-3637065 Answer to reviewer comments
We would like to sincerely thank the reviewers for their constructive comments and suggestions on how to improve our manuscript. A copy of the comments, along with a detailed reply, can be found below.
REPLY TO REVIEWER #1:
The manuscript by Xiang et al. explores the role of brain endothelial cells in the regulation of the blood–brain barrier (BBB), with a focus on therapeutic strategies targeting these cells in neurological diseases. While the topic is highly relevant and timely, the manuscript suffers from several major issues that need to be addressed. Despite an extensive number of citations, the discussion in many sections remains too superficial, lacking in mechanistic depth and experimental specificity. Furthermore, some statements are presented without appropriate referencing. A major revision is therefore strongly recommended.
Major comments:
Introduction (Lines 33–41):
- In line 33, I recommend the Authors list and briefly describe the key morphological and functional characteristics of brain endothelial cells (e.g., tight junctions, expression of transporters).
Answer: We have provided a brief description of the key morphological and functional characteristics of brain endothelial cells in lines 32–46 of the revised version.
- Lines 36–41 mention gene editing tools, but no specific examples are given. Please specify tools such as CRISPR/Cas9, or TALENs, and/or ZFNs, and briefly explain their relevance to BBB research.
Answer: We have added the applications of CRISPR/Cas9, TALENs, and ZFNs in the field of BEC research in lines 48–57 of the revised version.
- Notably, no references are provided in this entire section. Even for introductory content, appropriate citations are essential.
Answer: We have included reference citations in the introduction section.
- Cellular Origin of Brain Endothelial Cells:
- The section lacks methodological and experimental detail. For instance, when referring to developmental studies, it would be important to specify the use of zebrafish models, and describe the techniques used (e.g., transgenic lines, live imaging etc.).
Answer: Thanks for your suggestion. We have added experimental details in this section, including transgenic strains and experimental techniques.
- The conclusions presented are not sufficiently supported by detailed evidence. The section would benefit from greater clarity regarding how these insights were obtained.
Answer: Thanks for your comments. We have included experimental details to support the conclusions presented in this section.
- Morphology and Structure of Brain Endothelial Cells:
- Once again, this section lacks specific references to animal models. A review article should clearly state which systems (e.g., mice, rats, in vitro human-derived cells) underline the presented findings.
Answer: In this section, we have included the experimental models and specific experimental methods from the cited literature.
- Brain Endothelial Cells and Neurological Disorders:
- The discussion of disease models is too limited. The Authors should expand descriptions of individual disorders such as Alzheimer’s disease, gliomas, and stroke, and link them more clearly to BBB dysfunction.
Answer: We have expanded the characterization of pathological features in AD, stroke, and glioma, while enhancing the discussion on the association between these disease models and BBB dysregulation, particularly emphasizing the roles of VEGF and β-catenin in the pathogenesis of AD, MS, stroke, and glioma.
- Moreover, the Authors should elaborate on the specific studies that support their conclusions, including the models used and methods of evaluating endothelial pathology.
Answer: Thank you for your suggestion. We have incorporated additional references and employed experimental models to elucidate the molecular-level assessment of endothelial dysfunction in these pathologies.
- Therapeutic Strategies for Targeting Brain Endothelial Cells:
- This is the most underdeveloped section of the manuscript. The overview of drug delivery methods across the BBB is overly general and lacks a comprehensive analysis of current strategies.
Answer: Thanks for your comments. For each therapeutic strategy, we have incorporated a critical analysis of current methodological limitations, while delineating the potential roles of VEGF and β-catenin in modulating these treatment paradigms.
- The manuscript focuses mostly on PLGA nanoparticles, but omits discussion of other important delivery systems such as:
- Lipid-based nanoparticles, PEGylated systems, etc.;
- Viral vectors (e.g., AAV, lentiviruses) commonly used in gene therapy targeting the Central Nervous System;
- Exosomes, which are increasingly recognized as promising natural carriers due to their biocompatibility and ability to cross the BBB.
Answer: We systematically evaluate the applicability and mechanistic implications of diverse delivery systems—including lipid-based nanoparticles, PEGylated systems, viral vectors, and exosomes—in advancing therapeutic strategies targeting brain endothelial cells.
- The section should also address the challenges and translational barriers associated with these delivery systems, including immunogenicity, targeting specificity, and clinical feasibility.
Answer: Thank you for your suggestion. We have discussed the technical challenges and translational barriers inherent to these delivery systems.
Summary:
Although the manuscript tackles an important area of neurovascular research, the current version lacks sufficient scientific depth, methodological precision, and comprehensive coverage of key literature. Major improvements are needed in structure, clarity, and content—particularly regarding therapeutic approaches and mechanistic insights.
Answer: Thank you for your comments. We have expanded specific experimental models and methodological specifications through comprehensive revisions across all sections, with particular emphasis on substantially expanding the therapeutic strategies section to delineate the current advantages and persistent challenges of BEC-targeting therapeutic approaches.

Reviewer 2 Report
Comments and Suggestions for Authors
review ijms-3637065
Brain endothelial cells in blood-brain barrier regulation and neurological therapy
This review article deals with an interesting and thought-provoking view on brain endothelial cells, focusing in the first two chapters mainly on beta-catenin and VEGF, and the related protein metabolism.
Unfortunately, the focus on beta-catenin and VEGF is lost when dealing with the diseases in chapter 3, leading to more general assumptions and aleatory overviews. It would be most interesting to show the roles of beta-catenin and VEGF in the 4 neurologic disorders, with their effects on brain endothelial cells, and perhaps for the different treatment approaches. In the way presented here, the review is poorly structured and too short to be exhaustive.
A review focused on VEGF and beta-catenin through the five chapters might be of more interest to a scientific reader with an own background knowledge on BBB.
Minor points:
1. Abstract line 18. In a strict sense, neurotoxicity cannot infiltrate. Perhaps better: …, leading to neurotoxic effects in brain parenchyma and …
2. Line 58: chicks … better: avian embryos, or chick embryos
3. Line 74: a point at the end of the line, not a circle
4. Line 118: cues: perhaps better: … signals, … factors, … metabolism
5. Line 338: a point at the end of the line, not a circle
6. Line 360: a point at the end of the line, not a circle
7. Line 368: a point is missing between reference 116] and EPCs
Major points:
1. The authors do not describe, in how far the approaches of the different animal experiments are feasible for the use in human patients.
2. Why do lines 368-377 resemble so closely lines 378-386? It is not necessary to write it in double or to repeat citing of references ( EPCs are directly involved in neointima formation by secreting growth factors (e.g., vascular endothelial growth factor VEGF) and regulating pericyte activity through paracrine signaling to promote endothelial maturation and vascular repair [117]…)
3. The chapter 5. "Future perspectives" misses a concrete proposition or research idea based on the results of the review given here.
Figures
Figure 1: add lines 160 to 163 more clearly to the legend. Put ‘This figure was created in BioRender. https://BioRender.com’ into the references, with a number and “last accessed + date”. The legend should state: Schematic representation of two molecular regulation pathways involved in formation of blood-brain-barrier or fenestrated vasculature.
Figure 2: add lines 419 to 421 clearly to the legend.
Author Response
ijms-3637065 Answer to reviewer comments
We would like to sincerely thank the reviewers for their constructive comments and suggestions on how to improve our manuscript. A copy of the comments, along with a detailed reply, can be found below.
REPLY TO REVIEWER #2:
This review article deals with an interesting and thought-provoking view on brain endothelial cells, focusing in the first two chapters mainly on beta-catenin and VEGF, and the related protein metabolism.
Unfortunately, the focus on beta-catenin and VEGF is lost when dealing with the diseases in chapter 3, leading to more general assumptions and aleatory overviews. It would be most interesting to show the roles of beta-catenin and VEGF in the 4 neurologic disorders, with their effects on brain endothelial cells, and perhaps for the different treatment approaches. In the way presented here, the review is poorly structured and too short to be exhaustive.
A review focused on VEGF and beta-catenin through the five chapters might be of more interest to a scientific reader with an own background knowledge on BBB.
Answer: Thank you for your suggestion. In the revised manuscript, we have expanded the discussion to provide a comprehensive analysis of VEGF and β-catenin signaling in four neurological disorders, with a focus on their mechanistic impacts on BECs. Furthermore, we critically evaluate the therapeutic potential of these molecular targets in BEC-targeting strategies, addressing both current advancements and unresolved challenges.
Minor points:
1. Abstract line 18. In a strict sense, neurotoxicity cannot infiltrate. Perhaps better: …, leading to neurotoxic effects in brain parenchyma and …
Answer: We have revised the sentence into “…, leading to neurotoxic effects in brain parenchyma and …”
Line 58: chicks … better: avian embryos, or chick embryos
Answer: We have replaced the colloquial term “chicks” with the scientifically precise designation “avian embryos”.
Line 74: a point at the end of the line, not a circle
Answer: We have revised it.
Line 118: cues: perhaps better: … signals, … factors, … metabolism
Answer: We have replaced “cues” with “signals”.
Line 338: a point at the end of the line, not a circle
Answer: We have revised it.
Line 360: a point at the end of the line, not a circle
Answer: We have revised it.
Line 368: a point is missing between reference 116] and EPCs
Answer: We have added a point.
Major points:
1. The authors do not describe, in how far the approaches of the different animal experiments are feasible for the use in human patients.
Answer: We have analyzed the limitations of individual therapeutic approaches within the “BEC-targeting strategies” chapter and proposed evidence-based mitigation strategies. Additionally, the “Future Perspectives” section outlines actionable research initiatives to address unresolved challenges, including interspecies validation frameworks and the integration of human-relevant BBB-on-chip models.
Why do lines 368-377 resemble so closely lines 378-386? It is not necessary to write it in double or to repeat citing of references ( EPCs are directly involved in neointima formation by secreting growth factors (e.g., vascular endothelial growth factor VEGF) and regulating pericyte activity through paracrine signaling to promote endothelial maturation and vascular repair [117]…)
Answer: We have deleted the repetitive sentences.
The chapter 5. "Future perspectives" misses a concrete proposition or research idea based on the results of the review given here.
Answer: We have added related prospects in chatper5, including the treatment of diseases, the temporal-spatial regulation of endothelial cell signal transduction pathway and the crosstalk of cell types.
Figures
Figure 1: add lines 160 to 163 more clearly to the legend. Put ‘This figure was created in BioRender. https://BioRender.com’ into the references, with a number and “last accessed + date”. The legend should state: Schematic representation of two molecular regulation pathways involved in formation of blood-brain-barrier or fenestrated vasculature.
Answer: We have revised the figure legend and put “This figure was created in BioRender. https://BioRender.com” into the references, with a number and “last accessed + 13 April”
Figure 2: add lines 419 to 421 clearly to the legend.
Answer: We have revised the figure legend.

Round 2
Reviewer 1 Report
Comments and Suggestions for Authors
The Authors have improved the manuscript by incorporating more detailed information. However, there is still a lack of clarity regarding the specific animal model used.
1. In studies related to CNS disorders, it is particularly important to indicate the exact animal model. For instance, in subsection 2.2, line 217, the Authors refer to “adult mice,” but based on the referenced study, a more specific designation is necessary, such as “in adult male mice (C57BL/6J).” This clarification should be applied consistently throughout the manuscript.
2. Furthermore, in Figure 2, the name of the gene-editing system is incorrect. The proper term is CRISPR/Cas9. Please correct this in the Figure.
3. I also recommend adding a summary figure or table illustrating the described roles of the Wnt signaling pathway, anti-inflammatory factors, and anti-oxidative stress. Including such a schematic would enhance the clarity and readability of the manuscript.
Author Response
ijms-3637065 Answer to reviewer comments
We would like to sincerely thank the reviewers for their constructive comments and suggestions on improving our manuscript. A copy of the comments, followed by a detailed reply, can be found below.
REPLY TO REVIEWER #1:
The Authors have improved the manuscript by incorporating more detailed information. However, there is still a lack of clarity regarding the specific animal model used.
- In studies related to CNS disorders, it is particularly important to indicate the exact animal model. For instance, in subsection 2.2, line 217, the Authors refer to “adult mice,” but based on the referenced study, a more specific designation is necessary, such as “in adult male mice (C57BL/6J).”This clarification should be applied consistently throughout the manuscript.
Answer: We have revised it.
- Furthermore, in Figure 2, the name of the gene-editing system is incorrect. The proper term is CRISPR/Cas9. Please correct this in the Figure.
Answer: We have revised Figure 2. It should be noted that, due to the addition of a new figure, the original Figure 2 has been renumbered as Figure 3.
- I also recommend adding a summary figure or table illustrating the described roles of the Wnt signaling pathway, anti-inflammatory factors, and anti-oxidative stress. Including such a schematic would enhance the clarity and readability of the manuscript.
Answer: Thank you for your suggestion. A new schematic diagram (Figure 2) has been added to summarize the described roles of the Wnt signaling pathway, anti-inflammatory factors, and anti-oxidative stress, accompanied by a figure legend. The original Figure 2 has now been renamed as Figure 3.

Reviewer 2 Report
Comments and Suggestions for Authors
Brain endothelial cells in blood-brain barrier regulation and neurological therapy
The authors have carefully answered to the points and the review article is now easier to read. It is instructive and covers many interesting points.
In the way lines 358 to 360 are worded (see Major points 1 and 2), they do not correspond to the current scientific ideas.
Minor points:
- Line 139: … pericyte detachment in both retinal and ??? consequent exacerbation…, sentence incomplete
2. Line 142: … several, not severial
3. Line 151: … of plasma proteins, not plama
4. Line 165: remove ‘-‘ in plex-uses
5. Line 167: remove ‘mice’
6. Line 175: suppress one of the redundant references [33/34]
7. Line 217: no ‘,’ between murine and circumventricular
8. Line 218: no ‘-‘ between vascu-losum
9. Line 219: neuron and astrocyte-derived VEGF-A, not ‘as-trocyte-derived Vegf-A’
10. Line 656: suppress one of the redundant references [205, 206]
11. Line 796: suppress one of the redundant ‘in’s
12. Line 810: … of the BBB may render (skip: ‘make’) specific …
Major points:
1.Line 358: Glioblastoma, not ‘Glioma’, because there are many glioma subtypes.
2. Line 360: Notably … to tumor sites. Please review this sentence. The uptake of gadolinium in MRI of high-grade glioma is considered to be a sign of BBB-leakage. It might be more correct to formulate: “Notably, there are reports that the BBB in glioma can remain intact, and this preserved …”
3. The references 161 to 165 have not been cited in the text.
Author Response
ijms-3637065 Answer to reviewer comments
We would like to sincerely thank the reviewers for their constructive comments and suggestions on improving our manuscript. A copy of the comments, followed by a detailed reply, can be found below.
REPLY TO REVIEWER #2:
The authors have carefully answered to the points and the review article is now easier to read. It is instructive and covers many interesting points.
In the way lines 358 to 360 are worded (see Major points 1 and 2), they do not correspond to the current scientific ideas.
Minor points:
- Line 139: … pericyte detachment in both retinal and ??? consequent exacerbation…, sentence incomplete
Answer: We have revised this sentence.
Line 142: … several, not several
Answer: We have revised it.
Line 151: … of plasma proteins, not plama
Answer: We have revised it.
Line 165: remove ‘-‘ in plex-uses
Answer: We have revised it.
Line 167: remove ‘mice’
Answer: We have revised it.
Line 175: suppress one of the redundant references [33/34]
Answer: We have retained Reference [33].
Line 217: no ‘,’ between murine and circumventricular
Answer: We have revised it.
Line 218: no ‘-‘ between vascu-losum
Answer: We have revised it.
Line 219: neuron and astrocyte-derived VEGF-A, not ‘as-trocyte-derived Vegf-A’
Answer: We have revised it.
Line 656: suppress one of the redundant references [205, 206]
Answer: We have retained Reference [205]. In the current revision, this reference has been renumbered to [201].
Line 796: suppress one of the redundant ‘in’s
Answer: We have revised it.
Line 810: … of the BBB may render (skip: ‘make’) specific …
Answer: We have revised it.
Major points:
1.Line 358: Glioblastoma, not ‘Glioma’, because there are many glioma subtypes.
Answer: Thank you for your suggestion. We have replaced the “Glioma” with “Glioblastoma”.
Line 360: Notably … to tumor sites. Please review this sentence. The uptake of gadolinium in MRI of high-grade glioma is considered to be a sign of BBB-leakage. It might be more correct to formulate: “Notably, there are reports that the BBB in glioma can remain intact, and this preserved …”
Answer: Thank you for your comments. We have revised this sentence.
The references 161 to 165 have not been cited in the text.
Answer: We have revised it.

Round 3
Reviewer 1 Report
Comments and Suggestions for Authors
The Authors took into account the comments of the Reviewer and improved the manuscript significantly.
Author Response
The reviewer has no further questions.